# MultiMedEval: A Benchmark and a Toolkit for Evaluating Medical Vision-Language Models

**Corentin Royer**                                        CORENTIN.ROYER@UZH.CH
**Bjoern Menze**[*]                                       BJOERN.MENZE@UZH.CH
**Anjany Sekuboyina**[*]                                  ANJANY.SEKUBOYINA@UZH.CH
*University of Zürich*

**Editors:** Accepted for publication at MIDL 2024

## Abstract

We introduce MultiMedEval, an open-source toolkit for fair and reproducible evaluation of large, medical vision-language models (VLM). MultiMedEval comprehensively assesses the models' performance on a broad array of six multi-modal tasks, conducted over 23 datasets, and spanning over 11 medical domains. The chosen tasks and performance metrics are based on their widespread adoption in the community and their diversity, ensuring a thorough evaluation of the model's overall generalizability. We open-source a Python toolkit (https://github.com/corentin-ryr/MultiMedEval) with a simple interface and setup process, enabling the evaluation of any VLM in just a few lines of code. Our goal is to simplify the intricate landscape of VLM evaluation, thus promoting fair and uniform benchmarking of future models.

**Keywords:** Vision-Language Models, Medical, Multimodal, Benchmark, Toolkit.

## 1. Introduction

Large language models (LLM) and vision-language Models (VLM) are text generators capable of tackling a multitude of tasks based on textual or textual-and-visual prompts, *e.g.* question answering, machine translation, summarization, visual-question answering, image captioning, image classification, etc. Typically, assessing the performance of these models means evaluating them over a variety of tasks (mentioned above) on diverse datasets. This enables a reliable tracking of their progress and generalizability. General-purpose language models are therefore popularly benchmarked on toolkits such as OpenAI Evals, Huggingface LLM leaderboard (Beeching et al., 2023), and OpenVLM Leaderboard (Contributors, 2023). Such leaderboards offer a common platform for comparing open-access (Llama 2 (Touvron et al., 2023) and Flamingo (Alayrac et al., 2022)), and oftentimes, closed-source models (GPT-4V (Achiam et al., 2023) and Gemini (Team et al., 2023)) based on their performance.

Adapting VLMs to the medical domain proves challenging, primarily due to the domain-specific hurdles posed by proprietary datasets, fine-grained knowledge requirements, and the overall difficulty to generalize across medical domains and tasks. Despite these challenges, recent efforts culminated in truly capable *medical* VLMs. For instance, LLaVA-Med (Li et al., 2023), and PMC-VQA (Zhang et al., 2023b) build VLM assistants for medical VQA, while MAIRA-1 (Hyland et al., 2023) focuses on radiology report generation, specifically

---

[*] Joint supervision

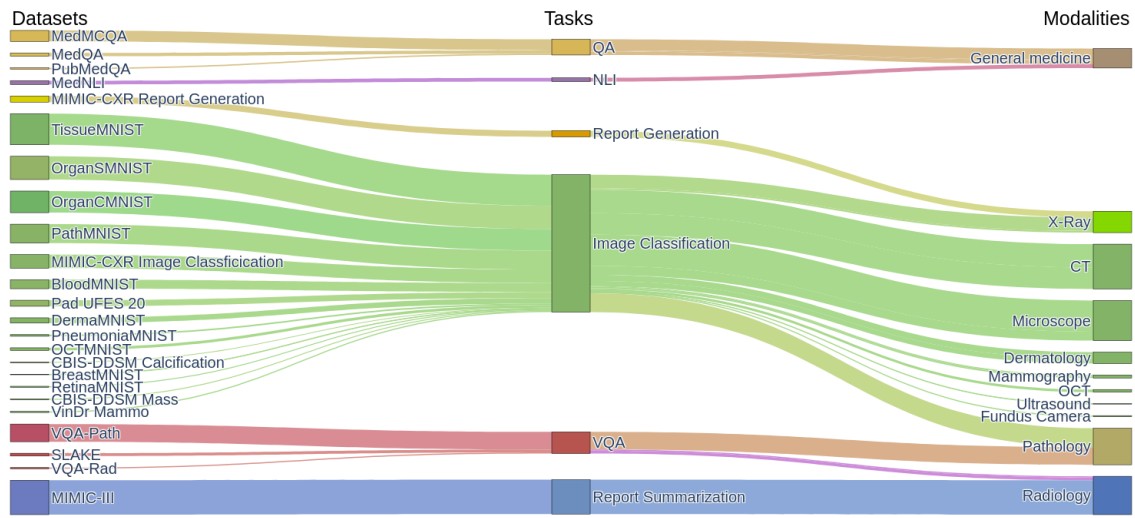

Figure 1: The left column represents the datasets in MultiMedEval and their size. Each dataset is used in a task, represented in the middle. Finally, we represent the share of each modality in the tool and the modality composition of each task.

chest x-rays (CXR). RadFM (Wu et al., 2023) proposes a versatile VLM, with a focus on radiology. Circumscribing the capabilities of the above-mentioned VLMs, MedPaLM M (Tu et al., 2023) and BiomedGPT (Zhang et al., 2023a) are pitched as *generalist* models capable of performing a wider array of tasks such as image classification, text summarization, etc.

Among these plethora of models, evaluation has been highly non-uniform. For instance, RadFM is evaluated on seven tasks while MedPaLM M is evaluated on five. Even among the tasks that both of them have been evaluated on, discrepancies exist in terms of either the datasets (for report generation MedPALM is evaluated on MIMI-CXR while RadFM is evaluated on three more datasets) or the metrics (BLEU and Recall in RadFM; BLEU and F1 in MedPALM). Similarly, LLAVA-Med reports six metrics, while RadFM reports 61 metrics; however, they share no common metrics, hindering a direct comparison of both approaches. Thus, there is a need for a unified benchmark, the lack of which has been consistently acknowledged (Wu et al., 2023; Li et al., 2023). Our work aims to build such a unified benchmark. Closely related to our work are the works by (Wu et al., 2023) and (Tu et al., 2023). However, the benchmark of (Wu et al., 2023) is specific to radiology-related tasks and doesn't include important domains such as general medicine. While the evaluation performed by (Tu et al., 2023) is holistic, it is closed-source, preventing a fair replication.

Summarizing the topic of the evaluation of medical VLMs, we identify three key issues that make evaluation difficult and eventually slow down medical VLM research. First, as stated above, models are benchmarked using different metrics, methodologies, and tasks preventing fair comparison to other models. Second, the scope of generalist models is so wide that every new model is required to benchmark against all the prior work. However,

re-implementing the metrics and recomputing benchmarks is time-consuming when models are open-access and impossible for closed-source models. Third, even if one plans to re-implement the benchmark, the evaluation pipelines (from data to metrics) are complex and cumbersome. Addressing these issues, **we introduce MultiMedEval, an open-source, Python-based evaluation toolkit for medical VLMs**. Our toolkit is designed to be user-friendly, offering reproducible evaluation capabilities across six distinct medical tasks implemented on 23 datasets spanning over 11 medical image and text modalities. Our benchmark encompasses a broader spectrum of medical domains than any model mentioned above. Additionally, we reimplemented two open-source models (RadFM and LLAVA-Med) and compared them with closed models according to the metrics reported therein. Our comprehensive comparison serves as a foundational baseline for future medical VLM research, with future evaluations seamlessly unified into this benchmark through MultiMedEval, resulting in its continued growth.

## 2. Tasks and Evaluations

The typical VLM is designed to take an interleaved image-text prompt as input and generate a textual response as output. For example, the prompt `<image> What is the modality of this image?` might generate a response `This image shows an MRI of the brain`. Any evaluation primarily focuses on the model's ability to respond to a variety of question prompts, which in turn defines the variety of *tasks* the model can perform. For a comprehensive evaluation of the VLM's capabilities, we evaluate it on six tasks: image classification, question answering (QA), visual QA, report summarization, report generation, and natural language inference (NLI). In every task, we detail the prompt design, datasets, and performance metrics that MultiMedEval employs during evaluation. We let the user decide if they want to do few-shot inference (*i.e.* prepend the prompt with examples of prompts and responses) or zero-shot inference. We use the official split for every dataset except MIMIC-III and Pad-UFES-20 which do not have an official split. We propose such a split for Pad-UFES 20 where we use 20% of the dataset for testing and we use the split proposed by (Delbrouck et al., 2022) for MIMIC-III. Fig. 1 gives an overview of the dataset, task, and modality distribution in MultiMedEval. For every task, example prompts for all the datasets are listed in the appendix.

**Multi-class and multi-label image classification.** We use a total of 15 datasets spanning nine modalities (*c.f.* Table 1). MIMIC-CXR is a multi-label classification task while the others are multi-class classification. The input prompt for image classification is constructed by the image followed by the classes and then a question, *e.g.* for OrganMNIST the full prompt will be: ` Options:1:bladder 2:femur-left ...  11:  spleen Which options correspond to the image?`.

For all datasets, except MIMIC-CXR, the model's predicted answer is determined by calculating the BLEU score between the model's response and each class, selecting the class with the highest score. For the MIMIC-CXR dataset, we use CheXBert (Smit et al., 2020) labeler on the response. CheXBert is a report-labeling tool that extracts 14 conditions of which we keep five conditions (Atelectasis, Cardiomegaly, Consolidation, Edema, and Pleural Effusion) for computing the metrics. Once the classes are extracted, we report the classification performance using macro F1, macro AUROC, and macro accuracy.

Table 1: List of the image classification datasets and the different modalities they cover.

| Modality | Dataset name | Classes | Size |
|---|---|---|---|
| CT (Radiology) | OrganSMNIST (Xu et al., 2019; Bilic et al., 2023) | 11 | 8827 |
| | OrganCMNIST (Xu et al., 2019; Bilic et al., 2023) | 11 | 8216 |
| Dermatology | Pad-UFES 20 (Pacheco et al., 2020) | 7 | 2298 |
| | DermaMNIST (Tschandl et al., 2018; Codella et al., 2019) | 7 | 2005 |
| Fundus Camera | RetinaMNIST (Liu et al., 2022) | 5 | 400 |
| Mammography | VinDr Mammo (Nguyen et al., 2023) | 5 | 429 |
| | CBIS-DDSM Mass (Lee et al., 2017) | 3 | 378 |
| | CBIS-DDSM Calcification (Lee et al., 2017) | 3 | 326 |
| Microscope | TissueMNIST (Ljosa et al., 2012) | 8 | 11820 |
| | BloodMNIST (Acevedo et al., 2020) | 8 | 3421 |
| OCT | OCTMNIST (Kermany et al., 2018) | 4 | 1000 |
| Pathology | PathMNIST (Kather et al., 2019) | 9 | 7180 |
| Ultrasound | BreastMNIST (Al-Dhabyani et al., 2020) | 2 | 7180 |
| X-Ray (Radiology) | PneumoniaMNIST (Kermany et al., 2018) | 2 | 7180 |
| | MIMIC Image Classification (Johnson et al., 2019) | 5 | 5159 |

Table 2: List of the QA datasets and the different modalities they cover.

| Modality | Dataset name | Size |
|---|---|---|
| General Medicine | MedQA (Lau et al., 2018) | 1273 |
| | MedMCQA (He et al., 2020) | 4183 |
| | PubMedQA (Liu et al., 2021) | 500 |

Table 3: List of the VQA datasets and the different modalities they cover.

| Modality | Dataset name | Size |
|---|---|---|
| Radiology | VQA-Rad (Lau et al., 2018) | 451 |
| | SLAKE (Liu et al., 2021) | 1061 |
| Pathology | Path-VQA (He et al., 2020) | 6719 |

**Question answering.** For QA, we evaluate on three datasets (*c.f.* Table 2): MedQA and MedMCQA consist of multi-choice questions (MCQ) and PubMedQA consists of close-ended questions (yes-no questions). For the MCQs, the model is prompted with the question followed by all the options and finally the phrase `What is the correct answer?`. For Pub-MedQA, the question is prepended with `Answer the question with yes, no, or maybe.`

To determine the model's predicted answer from its response, we utilize the BLEU metric (Papineni et al., 2002) to compare the predicted answer to each option, selecting the one with the highest BLEU score. For PubMedQA, we check if the answer contains the words `yes`, `no`, or `maybe`. Once the model's answers have been generated, we report answering performance using accuracy.

**Visual question answering.** For evaluating the performance of a VLM on VQA, we use three datasets (*c.f.* Table 3) containing a mix of open-ended and close-ended questions: Path-VQA, SLAKE, and VQA-Rad. The prompt for VQA is constructed by concatenating the image and the question, *e.g.* ` What is the main organ in the image?`. Similar to QA, the close-ended questions are prepended with `Answer the question with yes or no.`

Since there are no MCQs in VQA, the evaluation differs from QA. Specifically, the correct answer and the predicted one are tokenized and the resulting sets are used to compute precision and recall. We also differentiate between close-ended and open-ended questions to report close-ended accuracy, open-ended accuracy, and open-ended recall (Nguyen et al., 2019). Additionally, we also report overall recall and F1 score. Finally, we compute the BLEU score from the non-tokenized texts (Wu et al., 2023; Li et al., 2023). To calculate the accuracy, close-ended questions are considered correct if their recall is at least 0.5, while open-ended questions require a recall of at least 0.75.

**Report generation.** We include the MIMIC-CXR dataset (Johnson et al., 2019) containing de-identified radiology reports with the associated CXRs to evaluate report-generation capabilities. The task is to generate the *findings* section of the report based on the radi-

ology images. The input prompt for the model is constructed with the image (or multiple images pertaining to one case) followed by a sentence asking for the report, as in, `` ` Please caption this scan with findings and impressions.`

Following common practices, the generated reports are evaluated using n-gram-based methods: ROUGE-L, BLEU-1, BLEU-4, and METEOR (Banerjee and Lavie, 2005). Additionally, we compute F1-RadGraph, CheXBert vector similarity, F1-bertscore, and RadCliQ to capture the subtleties of radiological language. F1-RadGraph is the F1 score between the entities extracted from the reference report and generated one using RadGraph (Jain et al., 2021). F1-BertScore employs CheXBert to label the reference and generated reports, as introduced in the image classification task above. CheXBert vector similarity (Yu et al., 2023) computes the cosine similarity between the embedded reference and generated reports. Lastly, RadCliQ (Yu et al., 2023) is a composite metric composed of the previous four metrics, said to closely match the practitioners' feedback on report quality.

**Report summarization.** We evaluate report summarization on MIMIC-III (Johnson et al., 2016). Following (Van Veen et al., 2023), the VLM has to generate the *impressions* section of a radiology report based on the *findings* section. So, the model is prompted with the free-text findings (*e.g.* `Intracranial vessels are normal...  there is mild ven¨ triculomegal...  the subarachnoid hemorrhage noted in the right sylvian fissure`) followed by the task prompt, `Summarize the findings.` We use the same metric as for the report generation task to compare the ground truth impressions to the generated summary.

**Natural language inference.** Gauging the logical reasoning capabilities of the VLM, Natural language inference (NLI) involves the categorization of pairs of sentences into three classes: contradiction, entailment, or neutral, effectively resembling a 3-class classification task. We evaluate NLI using MedNLI (Romanov and Shivade, 2018), a dataset consisting of pairs of medical statements. We prompt the model with the two sentences from a pair followed by a question asking to classify the logical relationship between them, as in `<sentence-1> <sentence-2> Determine the logical relationship between these two sentences.`

To extract the VLM's predicted answer from its response, we check the presence of either of the three terms (contradiction, entailment, or neutral). For the answer to be valid, only one of the three classes must be present; if none of the three or several of them are present, the answer is deemed invalid. The performance of the model is then reported using accuracy.

## 3. MultiMedEval Setup and Utilization

Ideally, once a VLM is developed, the entire suite of evaluations mentioned above needs to be conducted. Typically, this involves downloading the datasets, implementing the data pre-processing, implementing the computation of the metrics, running inference through the VLM, and finally recording the performance. We design MultiMedEval to abstract this entire pipeline, only exposing APIs for setting up the data and for evaluating on them. In this section, we briefly describe the usage of MultiMedEval and strongly encourage the reader to peruse the official documentation (currently hosted at https://github.com/corentin-ryr/MultiMedEval).

**Parameters.** MultiMedEval exposes two parameter classes, `SetupParams` and `EvalParams`. The setup parameters control the data download. For the datasets hosted on PhysioNet, the setup process also requires appropriate credentials. The evaluation parameters control the evaluation configuration such as `batch_size`, `device` (GPU-id), etc.

**Batcher.** The only code that the user needs to implement is a Callable, `batcher`, which wraps around the user's VLM inference module. Every call to `batcher` takes, as input, a batch of conversation prompts and returns, as output, the decoded model responses. The input prompts are constructed according to HuggingFace's conversation style.

**Evaluation.** Once the parameters and the `batcher` are ready, the evaluation can be run using the `eval` API, which takes as input the list of datasets that the user wants to benchmark on, along with the `batcher` and the evaluation parameters. The results of the evaluation are saved as a JSON file. Below, we provide the pseudocode for one such evaluation:

```python
from multimedeval import MultiMedEval, SetupParams, EvalParams

# Implementing the batcher for the user's specific model and returning text
    answers
def exampleBatcher(prompts:list[tuple]) -> list[str]:
    return [model.generate(prompt) for prompt in prompts]

engine = MultiMedEval()
# Running the setup only for MedQA
setupParams = SetupParams(MedQA_dir="data/")
engine.setup(setupParams)

# Running the evaluation on the exampleBatcher
evalParams = EvalParams(batch_size=32)
engine.eval(["MedQA"], exampleBatcher, evalParams)
```

## 4. Baselines

As previously indicated, MultiMedEval's purpose is to enable a comprehensive assessment of any VLM. To demonstrate this, we benchmark two recent, publicly-available models, RadFM (Wu et al., 2023) and LLaVA-Med (Li et al., 2023). In this benchmark, we also include the performance reported by two closed models, MedPALM M (Tu et al., 2023) and Maira-1 (Hyland et al., 2023), as well as one very recent public model, BiomedGPT (Zhang et al., 2023a). In Tables 4 and 5, we report the complete picture of the model performances across six tasks, 23 datasets, and 81 metrics. The performance is grouped by tasks and color-coded in green. The brighter the color, the superior the performance (*i.e.* highest accuracy or lowest RadCliQ score is the brightest). A gray cell indicates that the performance was never reported.

Owing to a holistic picture provided by MultiMedEval, we make five crucial observations: First, there is not a single task or metric that every medical VLM, to date, has been

Table 4: Performance of baseline VLMs on MultiMedEval's tasks. Brighter the cell, better the performance. Grey values indicate metrics that the model was not evaluated with. Table continued in Table 5.

| Dataset | Metric | RadFM | RaFM (reported) | LLaVA-Med | LLaVA-Med (reported) | Med-PaLM M (reported) | Maira-1 (reported) | Biomedgpt (reported) |
|---|---|---|---|---|---|---|---|---|
| Image classification | | | | | | | | |
| MIMIC-CXR Image classification | Macro-AUC↑ | 0.536 | - | 0.470 | - | 0.791 | - | - |
| | Accuracy↑ | 0.728 | - | 0.668 | - | - | - | 0.897 |
| | Macro-F1↑ | 0.244 | - | 0.097 | - | 0.416 | - | - |
| PAD-UFES-20 | Macro-AUC↑ | 0.428 | - | 0.434 | - | 0.973 | - | - |
| | Accuracy↑ | 0.141 | - | 0.148 | - | - | - | - |
| | Macro-F1↑ | 0.093 | - | 0.132 | - | 0.843 | - | - |
| VinDr-Mammo | Macro-AUC↑ | 0.300 | - | 0.300 | - | 0.718 | - | - |
| | Accuracy↑ | 0.333 | - | 0.193 | - | - | - | - |
| | Macro-F1↑ | 0.069 | - | 0.190 | - | 0.357 | - | - |
| CBIS-DDSM-Mass | Macro-AUC↑ | 0.535 | - | 0.498 | - | 0.733 | - | - |
| | Accuracy↑ | 0.397 | - | 0.332 | - | - | - | - |
| | Macro-F1↑ | 0.258 | - | 0.090 | - | 0.511 | - | 0.572 |
| CBIS-DDSM-Calc | Macro-AUC↑ | 0.445 | - | 0.500 | - | 0.822 | - | - |
| | Accuracy↑ | 0.250 | - | 0.333 | - | - | - | - |
| | Macro-F1↑ | 0.169 | - | 0.123 | - | 0.679 | - | 0.728 |
| MNIST-Oct | Macro-AUC↑ | 0.500 | - | 0.502 | - | - | - | - |
| | Accuracy↑ | 0.250 | - | 0.252 | - | - | - | 0.816 |
| | Macro-F1↑ | 0.100 | - | 0.148 | - | - | - | - |
| MNIST-Path | Macro-AUC↑ | 0.500 | - | 0.485 | - | - | - | - |
| | Accuracy↑ | 0.111 | - | 0.086 | - | - | - | 0.926 |
| | Macro-F1↑ | 0.019 | - | 0.060 | - | - | - | - |
| MNIST-Blood | Macro-AUC↑ | 0.501 | - | 0.500 | - | - | - | - |
| | Accuracy↑ | 0.126 | - | 0.125 | - | - | - | 0.977 |
| | Macro-F1↑ | 0.041 | - | 0.055 | - | - | - | - |
| MNIST-Breast | Macro-AUC↑ | 0.500 | - | 0.476 | - | - | - | - |
| | Accuracy↑ | 0.500 | - | 0.476 | - | - | - | 0.878 |
| | Macro-F1↑ | 0.212 | - | 0.454 | - | - | - | - |
| MNIST-Derma | Macro-AUC↑ | 0.505 | - | 0.500 | - | - | - | - |
| | Accuracy↑ | 0.153 | - | 0.144 | - | - | - | 0.786 |
| | Macro-F1↑ | 0.015 | - | 0.019 | - | - | - | - |
| MNIST-OrganC | Macro-AUC↑ | 0.500 | - | 0.497 | - | - | - | - |
| | Accuracy↑ | 0.090 | - | 0.086 | - | - | - | 0.931 |
| | Macro-F1↑ | 0.009 | - | 0.024 | - | - | - | - |
| MNIST-OrganS | Macro-AUC↑ | 0.500 | - | 0.501 | - | - | - | - |
| | Accuracy↑ | 0.090 | - | 0.092 | - | - | - | 0.823 |
| | Macro-F1↑ | 0.008 | - | 0.026 | - | - | - | - |
| MNIST-Pneumonia | Macro-AUC↑ | 0.500 | - | 0.457 | - | - | - | - |
| | Accuracy↑ | 0.500 | - | 0.457 | - | - | - | 0.967 |
| | Macro-F1↑ | 0.272 | - | 0.413 | - | - | - | - |
| MNIST-Retina | Macro-AUC↑ | 0.500 | - | 0.487 | - | - | - | - |
| | Accuracy↑ | 0.200 | - | 0.178 | - | - | - | - |
| | Macro-F1↑ | 0.121 | - | 0.082 | - | - | - | - |
| MNIST-Tissue | Macro-AUC↑ | 0.500 | - | 0.500 | - | - | - | - |
| | Accuracy↑ | 0.125 | - | 0.126 | - | - | - | - |
| | Macro-F1↑ | 0.021 | - | 0.081 | - | - | - | - |
| NLI | | | | | | | | |
| MedNLI | Accuracy↑ | 0.001 | - | 0.080 | - | - | - | 0.838 |

evaluated on. This speaks to the non-uniformity in the existing evaluation regimes. Second, closed models such as MAIRA-1 (on report generation) and MedPaLM M (on QA, image classification, etc.) show superior performance compared to open-source models. Third, we can see an improvement in VLMs' performance on QA and image classification. In both cases, MedPaLM M outperforms the others by a significant margin. Fourth, the most recent open-source model (BiomedGPT) seems to be very promising. On metrics reported by both MedPaLM M and BiomedGPT, the latter shows a competitive performance. At image classification (Macro-F1 on CBIS-DDSM), BiomedGPT even outperforms MedPaLM M, showing an encouraging prospect for open-source models. Fifth and finally, the number of empty cells clearly showcases the need for a standardized evaluation protocol for medical

Table 5: Continuation of the VLM benchmark from Table 4.

| Dataset | Metric | RadFM | RaFM (reported) | LLaVA-Med | LLaVA-Med (reported) | Med-PaLM M (reported) | Maira-1 (reported) | Biomedgpt (reported) |
|---|---|---|---|---|---|---|---|---|
| | | | | QA | | | | |
| MedQA | Accuracy↑ | 0.230 | - | 0.241 | - | 0.697 | - | - |
| MedMCQA | Accuracy↑ | 0.288 | - | 0.309 | - | 0.626 | - | - |
| PubMedQA | Accuracy↑ | 0.336 | - | 0.488 | - | 0.800 | - | - |
| | | | Report summarization | | | | | |
| MIMIC-III | ROUGE-L↑ | 0.157 | - | 0.164 | - | 0.320 | - | 0.307 |
| | ROUGE-1↑ | 0.221 | - | 0.222 | - | - | - | - |
| | BLEU-1↑ | 0.127 | - | 0.129 | - | 0.154 | - | - |
| | BLEU-4↑ | 0.016 | - | 0.030 | - | - | - | - |
| | F1-RadGraph↑ | 0.149 | - | 0.211 | - | 0.347 | - | 0.312 |
| | RadCliQ↓ | 1.396 | - | 1.315 | - | - | - | - |
| | CheXbert vector↑ | 0.602 | - | 0.596 | - | - | - | - |
| | METEOR↑ | 0.191 | - | 0.314 | - | - | - | - |
| | | | | VQA | | | | |
| VQA-RAD | BLEU-1↑ | 0.443 | 0.522 | 0.058 | - | 0.713 | - | - |
| | closed Q accuracy↑ | 0.581 | - | 0.621 | 0.614 | - | - | 0.732 |
| | open Q recall↑ | 0.326 | - | 0.335 | 0.282 | - | - | - |
| | recall↑ | 0.468 | 0.428 | 0.485 | - | - | - | - |
| | open Q accuracy↑ | 0.260 | - | 0.255 | - | - | - | 0.732 |
| | F1↑ | 0.454 | - | 0.102 | - | 0.621 | - | - |
| Slake-VQA | BLEU-1↑ | 0.707 | 0.786 | 0.051 | - | 0.927 | - | - |
| | closed Q accuracy↑ | 0.715 | - | 0.515 | 0.522 | - | - | 0.861 |
| | open Q recall↑ | 0.720 | - | 0.408 | 0.392 | - | - | - |
| | recall↑ | 0.718 | 0.744 | 0.444 | - | - | - | - |
| | open Q accuracy↑ | 0.684 | - | 0.361 | - | - | - | 0.861 |
| | F1↑ | 0.718 | - | 0.090 | - | 0.893 | - | - |
| Path-VQA | BLEU-1↑ | 0.254 | - | 0.029 | - | 0.723 | - | - |
| | closed Q accuracy↑ | 0.500 | - | 0.511 | 0.541 | - | - | 0.581 |
| | open Q recall↑ | 0.019 | - | 0.079 | 0.123 | - | - | - |
| | recall↑ | 0.259 | - | 0.290 | - | - | - | - |
| | open Q accuracy↑ | 0.008 | - | 0.037 | - | - | - | 0.581 |
| | F1↑ | 0.256 | - | 0.052 | - | 0.627 | - | - |
| | | | Report generation | | | | | |
| MIMIC-CXR Report Generation | F1-RadGraph↑ | 0.108 | - | 0.021 | - | 0.267 | 0.243 | - |
| | BLEU-1↑ | 0.127 | 0.194 | 0.109 | - | 0.323 | 0.392 | - |
| | BLEU-4↑ | 0.004 | - | 0.000 | - | 0.115 | 0.142 | - |
| | ROUGE-1↑ | 0.208 | 0.262 | 0.149 | - | - | - | - |
| | ROUGE-L↑ | 0.128 | - | 0.103 | - | 0.275 | 0.289 | - |
| | RadCliQ↓ | 1.994 | - | 2.415 | - | - | 3.100 | - |
| | CheXbert vector↑ | 0.214 | - | 0.156 | - | - | 0.440 | - |
| | METEOR↑ | 0.178 | - | 0.123 | - | - | 0.333 | - |

VLMs as well as an easy-to-use toolkit that performs the evaluation so that researchers can cater more focus on the development of the models.

## 5. Conclusion

Medical vision-language models are just gathering momentum, they already show interesting generalizable capabilities, and their capabilities are bound to expand. Therefore, this is an opportune moment to establish a standard evaluation protocol based on community consensus. Addressing this, we presented MultiMedEval, a Python toolkit to comprehensively assess the performance of any VLM model on multiple medical tasks. Using this, we benchmarked RadFM and LLaVVa-Med and compared their results to the reported performances of state-of-the-art medical VLMs.

**Future work.** MultiMedEval will be released to the community and will be actively maintained by adding new tasks, metrics, and datasets. To this end, we will work with open-source medical imaging libraries such as MONAI (MONAI Consortium, 2023) and MLCommons (MLCommons Consortium) to increase community adoption.

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

## Appendix A. Example of prompts

We give examples of prompts for each of the datasets that we benchmark in our tool. For each example, we put the text from the dataset in black and the prompt we added in gray.

Table A1: Prompt example for each of the MedMNIST image classification datasets.

| Dataset | Prompt example |
|---|---|
| OCT MNIST | ` Options: \n 1: choroidal neovascularization \n 2: diabetic macular edema \n 3: drusen \n 4: normal \n` Which options correspond to the image? |
| Path MNIST | ` Options:\n 1: adipose \n 2: background \n 3: debris \n 4: lym¨ phocytes \n 5: mucus \n 6: smooth muscle \n 7: normal colon mucosa \n 8: cancer-associated stroma \n 9: colorectal adenocarcinoma epithelium \n` Which options correspond to the image? |
| Blood MNIST | ` Options:\n 1: basophil \n 2: eosinophil \n 3: erythroblast \n 4: immature granulocytes(myelocytes, metamyelocytes and promyelocytes) \n 5: lymphocyte \n 6: monocyte \n 7: neutrophil \n 8: platelet \n` Which options correspond to the image? |
| Breast MNIST | ` Options:\n 1: malignant \n 2: normal, benign \n` Which options correspond to the image? |
| Derma MNIST | ` Options:\n 1: actinic keratoses and intraepithelial carcinoma \n 2: basal cell carcinoma \n 3: benign keratosis-like lesions \n 4: dermatofi¨ broma \n 5: melanoma \n 6: melanocytic nevi \n 7: vascular lesions \n` Which options correspond to the image? |
| OrganC MNIST | ` Options:\n 1: bladder \n 2: femur-left \n 3: femur-right \n 4: heart \n 5: kidney-left \n 6: kidney-right \n 7: liver \n 8: lung-left \n 9: lung-right \n 10: pancreas \n 11: spleen \n` Which options correspond to the image? |
| OrganS MNIST | ` Options:\n 1: bladder \n 2: femur-left \n 3: femur-right \n 4: heart \n 5: kidney-left \n 6: kidney-right \n 7: liver \n 8: lung-left \n 9: lung-right \n 10: pancreas \n 11: spleen \n` Which options correspond to the image? |
| Pneumonia MNIST | ` Options:\n 1: normal \n 2: pneumonia \n` Which options correspond to the image? |
| Retina MNIST | ` Options:\n 1: 0 \n 2: 1 \n 3: 2 \n 4: 3 \n 5: 4 \n` Which options correspond to the image? |
| Tissue MNIST | ` Options:\n 1: Collecting Duct, Connecting Tubule \n 2: Distal Convo¨ luted Tubule \n 3: Glomerular endothelial cells \n 4: Interstitial endothe¨ lial cells \n 5: Leukocytes \n 6: Podocytes \n 7: Proximal Tubule Segments \n 8: Thick Ascending Limb \n` Which options correspond to the image? |

Table A2: Prompt example for each of the image classification datasets.

| Dataset | Prompt example |
|---|---|
| MIMIC-CXR Image Classification |  List the conditions that can be seen in this picture. |
| VinDr Mammo |  What is the BI-RADS level in this mammography (from 1 to 5)? |
| Pad UFES 20 |  Options: Basal Cell Carcinoma (BCC) Squamous Cell Carcinoma (SCC) Actinic Keratosis (ACK) Seborrheic Keratosis (SEK) Bowen's dis¨ease (BOD) Melanoma (MEL) Nevus (NEV) What is the most likely diagno¨sis among the following propositions? |
| CBIS-DDSM Mass |  Is the mass benign, malignant or benign without callback? |
| CBIS-DDSM Calcification |  Is the calcification benign, malignant or benign without call¨back? |

Table A3: Prompt example for each of the QA datasets.

| Dataset | Prompt example |
|---|---|
| MedQA | A 67-year-old man with transitional cell carcinoma of the bladder comes to the physician because of a 2-day history of ringing sensation in his ear. He received this first course of neoadjuvant chemotherapy 1 week ago. Pure tone audiometry shows a sensorineural hearing loss of 45 dB. The expected benefi¨ cial effect of the drug that caused this patient's symptoms is most likely due to which of the following actions? Options: A: Inhibition of thymidine syn¨ thesis. B: Inhibition of proteasome. C: Hyperstabilization of microtubules. D: Generation of free radicals. E: Cross-linking of DNA. What is the correct answer? |
| MedMCQA | Which of the following is not true for myelinated nerve fibers: a: Impulse through myelinated fibers is slower than non-myelinated fibers. b: Membrane currents are generated at nodes of Ranvier. c: Saltatory conduction of im¨ pulses is seen. d: Local anesthesia is effective only when the nerve is not covered by myelin sheath. What is the correct answer? |
| PubMedQA | Answer the question with yes, no or maybe. Dyschesia can be provoked by in¨ appropriate defecation movements. The aim of this prospective study was to demonstrate dysfunction of the anal sphincter and/or the musculus (m.) pub¨ orectalis in patients with dyschesia using anorectal endosonography. Twenty consecutive patients with a medical history of dyschesia and a control group of 20 healthy subjects underwent linear anorectal endosonography (Toshiba models IUV 5060 and PVL-625 RT). In both groups, the dimensions of the anal sphincter and the m. puborectalis were measured at rest, and during voluntary squeezing and straining. Statistical analysis was performed within and be¨ tween the two groups. The anal sphincter became paradoxically shorter and/or thicker during straining (versus the resting state) in 85% of patients but in only 35% of control subjects. Changes in sphincter length were statisti¨ cally significantly different (p<0.01, chi(2) test) in patients compared with control subjects. The m. puborectalis became paradoxically shorter and/or thicker during straining in 80% of patients but in only 30% of controls. Both the changes in length and thickness of the m. puborectalis were signifi¨ cantly different (p<0.01, chi(2) test) in patients versus control subjects. Is anorectal endosonography valuable in dyschesia? |

Table A4: Prompt example for each of the VQA datasets.

| Dataset | Prompt example |
|---|---|
| VQA-Rad | Answer the following question with yes or no.  are regions of the brain infarcted? |
| Path-VQA |  where are liver stem cells (oval cells) located? |
| SLAKE |  What is the main organ in the image? |

Table A5: Prompt example for report generation and report summarization.

| Dataset | Prompt example |
|---------|----------------|
| MIMIC-III | intracranial vessels are all normal in appearance including the carotid ar¨ teries and circle of ⎵⎵without any aneurysm identified.  again noted is the right parafalcine subdural hematoma extending over the right tentorium with¨ out significant change from recent prior exam.  no gross reaccumulation of the left sided subdural hematoma is appreciated, though evaluation for subtle hemorrhage is difficult as this was a contrast-enhanced study.  there has been interval placement of a right sided ventriculostomy tube terminating in the posterior aspect of the right lateral ventricle.  again noted is some high attenuation material in the left lateral ventricle, third ventricle and fourth ventricle which likely represents clot adherent to choroid plexus.  there is mild ventriculomegaly which is not significantly changed from the prior exam. intraparenchymal high attenuation surrounding the ventriculostomy tube in the right frontal lobe likely represents a small amount of intraparenchymal hem¨ orrhage.  the subarachnoid hemorrhage noted in the right sylvian fissure on the prior exam is difficult to appreciate due to contrast-enhancement, but is likely not significantly changed.  limited views through the cervical spine demonstrate multilevel spondylosis with mild anterolisthesis of c3 upon c4, likely on the basis of facet degenerative change as no spondylolysis is iden¨ tified.  there is no significant central canal stenosis.  incidentally noted is enlargement of the right lobe of the thyroid gland which contains a fo¨ cus of high attenuation possibly representing calcification. Summarize the findings. |
| MIMIC-CXR | \ \ \ Please caption this scan with findings and impression. |

Table A6: Prompt example for each of the Natural Language Inference datasets.

| Dataset | Prompt example |
|---------|----------------|
| MedNLI | Sentence 1:  Labs were notable for Cr 1.7 (baseline 0.5 per old records) and lactate 2.4.\n Sentence 2:  Patient has elevated Cr\n Determine the logical relationship between these two sentences.Does the second sentence logically follow from the first (entailment), con¨ tradict the first (contradiction), or if there is no clear logical relationship between them (neutral)? |

Table A7: Origin of the prompt for each dataset. For the first 8, we took the same prompt as the reference paper. For the other ones, we adapted or created a new prompt entirely. We give examples of the adapted prompts to show the differences.

| Dataset | Reference |
| --- | --- |
| MedQA | Med-PALM M |
| MedMCQA | Med-PALM M |
| PubMedQA | Med-PALM M |
| VQA-Rad | LLaVA-Med |
| SLAKE | LLaVA-Med |
| VQA-Path | LLaVA-Med |
| MIMIC-CXR Report gen | MAIRA-1 & RadFM |
| Pad UFES 20 | Med-PALM M |
| VinDr Mammo | Med-PALM M: Given mammogram image . Image view: bilateral craniocaudal Q: What is the most likely breast BI-RADS score? (A) 1 (B) 2 (C) 3 (D) 4 (E) 5
MultiMedEval: What is the BI-RADS level in this mammography (from 1 to 5)? |
| CBIS-DDSM Mass | Med-PALM M: Given mammogram image . Image view: CC Q: Which of the following is the most likely type of the patient's breast calcification? (A) BENIGN (B) BENIGN_WITHOUT_CALLBACK (C) MALIGNANT
MultiMedEval: Is the mass benign, malignant or benign without callback? |
| CBIS-DDSM Calcification | Med-PALM M: Given mammogram image . Image view: CC Q: Which of the following is the most likely type of the patient's breast calcification? (A) BENIGN (B) BENIGN_WITHOUT_CALLBACK (C) MALIGNANT
MultiMedEval: Is the mass benign, malignant or benign without callback? |
| MIMIC-CXR Image classification | Med-PALM M: Identify if a specific type of abnormality is shown in the X-ray. Given the AP view X-ray image . Q: Is cardiomegaly indicated by the image? (A) No (B) Yes
MultiMedEval: List the conditions that can be seen in this picture. |
| All MedMNIST datasets & MedNLI | BiomedGPT does not mention the prompt they use so used the prompts listed in Table A. |

## Appendix B. Text cleaning and tokenization for VQA tasks

For the VQA tasks, the generated answer is cleaned and tokenized to compute the recall which is in turn used for the accuracy metrics. This preprocessing pipeline is in line with the one used in LLaVA-Med. First, we clean the generated string by removing punctuation and articles ("a", "an", "the"), then we turn all numbers into numerals ("one" → "1") and expand contractions that lack apostrophes. We then split the sentence at spaces and end up with a list of words that we turn into a set (removing duplicates).

