# OpenReview forum: "MultiMedEval: A Benchmark and a Toolkit for Evaluating Medical Vision-Language Models"
_MIDL.io/2024/Conference — MIDL 2024 Poster_

### Official Review · Reviewer_76KA · 2024-02-22

**Confidence:** 4
**Preliminary Rating:** 4
**Recommendation:** Poster
**Final Rating:** 5

**Summary:**

This paper introduces MultiMedEval, an open-source toolkit for fair and reproducible evaluation of large, medical vision-language models (VLM). MultiMedEval comprehensively assesses the models’ performance on a broad array of six multi-modal tasks, conducted over 23
datasets, and spanning over 11 medical domains. This is a meaningful endeavor that requires sustained long-term efforts for maintenance and success.

**Strengths:**

1. MultiMedEval comprehensively assesses the models’ performance on a broad array of six multi-modal tasks, conducted over 23
datasets, and spanning over 11 medical domains.
2. They introduce an open-source toolkit for fair and reproducible evaluation of medical VLM.

**Weaknesses:**

1. diff-VQA[1] is missing.
[1] Expert Knowledge-Aware Image Difference Graph Representation Learning for Difference-Aware Medical Visual Question Answering
2. Introducing additional fairness evaluation metrics is important. The authors have not addressed the fairness of large models and its metric assessment.

**Detailed Comments:**

1. Diff-VQA[1] should be incorporated for evaluation.
[1] Expert Knowledge-Aware Image Difference Graph Representation Learning for Difference-Aware Medical Visual Question Answering
2. The importance of long-term maintenance should not be underestimated. With the rapid advancements in the field of Vision-Language Models (VLM), it is a critical question whether the authors are willing and able to continuously update the model to keep up with emerging ones, as this might require significant time and effort.
3. In assessing the fairness of existing medical VLMs, it is recommended to introduce additional metrics. For instance, one could focus on examining the model's performance across diverse demographics, types of diseases, or sources of medical images. Additionally, incorporating fairness metrics such as balance, equality, and disparity can help evaluate whether the model's performance is equitable across different subgroups, ensuring fairness under various circumstances. How do the authors address fairness in large models and evaluate it through specific metrics?

**Justification Of Final Rating:**

Thanks to the author's positive improvements, I believe several issues that were concerning to me have been addressed.
They added Diff-VQA to MultiMedEval.
They are concerning fairness.
As a result, I've decided to adjust my rating accordingly.

**Justification Of The Preliminary Rating:**

The criteria for evaluation stem from a comprehensive examination of the motivation and innovative aspects underlying the visual-language pretrained model. This involves a thorough exploration of the rationale behind the model's development and the unique contributions it brings to the field. Furthermore, the evaluation encompasses a comparative analysis of the model's experimental results against those of state-of-the-art methods, aiming to assess its performance and potential advancements over existing approaches.

**Questions To Address In The Rebuttal:**

Taking into account the aforementioned weaknesses and detailed comments,

**Special Issue:**

No

---

> ### Author Response · Authors · 2024-03-16
>
> We appreciate your review and we are glad you recognize the scope and work that went into MultiMedEval. We tried to address your concerns.
>
>
> ## Concern 1: Addition of Diff-VQA
>
>
> Thank you for pointing us towards Diff-VQA. We have now added Diff-VQA to MultiMedEval [Diff-VQA pull request](https://github.com/corentin-ryr/MultiMedEval/pull/4). The full evaluation is currently underway and the results will be updated on GitHub.
>
>
> Your comment also gave us an idea for a new feature: to allow the user to add any new dataset. Just format the dataset as a JSON file and treat it as one of the task “families” of MultiMedEval (VQA, QA, Report Comparison, Image classification). The tool will then apply the same metrics as the other tasks in the "family" (e.g. accuracy for QA, BLEU, accuracy, recall for VQA). More details can be found on the GitHub repository.
>
>
> ## Concern 2: Stratified Analysis and Fairness
>
>
>
>
> This is indeed good feedback, the option to perform a more detailed evaluation per task. Especially concerning fairness, we are contemplating analysis of existing datasets (Pad-UFES 20) as well as using additional datasets (MIMIC and Genderbias_CheXnet; https://arxiv.org/pdf/2310.09507.pdf). Given that this is a longer exercise, we track the progress on this issue: [GitHub issue](https://github.com/corentin-ryr/MultiMedEval/issues/6). Any other ideas in this regard are more than welcome as feature requests on the repository.
>
>
> ## Concern 3: Long-term maintenance
>
>
> We fully agree with you that such a tool is as good as it is maintained. In the near term, the tool will be actively maintained, as evidenced by the commits performed during this MIDL review process. Simultaneously, we plan to develop features that will eventually render large-scale development unnecessary (e.g. possibility of using any new dataset). Simultaneously, once the tool is stable, we plan to reach out to open-source libraries such as MONAI and MLCommons and transition the maintenance to the community.

---

### Official Review · Reviewer_iVsF · 2024-02-28

**Confidence:** 4
**Preliminary Rating:** 3
**Recommendation:** Poster
**Final Rating:** 4

**Summary:**

The paper introduces an open-source benchmark toolkit, namely MultiMedEval, for fair and comprehensive evaluation of medical vision-language models (VLM). The benchmark consists of six multi-modal tasks with 23 datasets, spanning 11 medical domains. Two open-source models (RadFM and LLAVA-Med) are evaluated using MultiMedEval and compared with their reported performances, as well as the reported performances of three closed models (MedPaLM M, MAIRA-1, and BiomedGPT). MultiMedEval establishes a standard evaluation protocol for the community.

**Strengths:**

1. The paper presents a comprehensive benchmark for evaluating medical vision-language models.
2. MultiMedEval employs a diverse set of six multi-modal tasks spanning 23 datasets across 11 medical domains to evaluate the model's efficacy.
3. The paper elucidates the rationale behind selecting metrics for each task, grounded in community consensus and accompanied by justifications.

**Weaknesses:**

1. The primary concern regarding the proposed open-source toolkit is its reliability. Tables 4 and 5 reveal significant disparities between the performance scores of RadFM and LLAVA-Med evaluated by MultiMedEval and their reported scores. For instance, in the MIMIC-CXR Report Generation task, MultiMedEval yields considerably lower BLEU-1 and ROUGE-L scores (0.006 and 0.065, respectively) compared to the reported scores of 0.194 and 0.262. Similarly, in image classification tasks, a majority of the Macro-AUC and Accuracy scores are either equal to or less than 0.5, casting doubt on the credibility and reliability of MultiMedEval.

2. The paper's explanation of the evaluation method for visual question answering lacks clarity. It mentions that "the correct answer and the predicted one are tokenized, and the resulting sets are used to compute precision and recall." However, it is essential to clarify how the answers are tokenized and which model is employed for this tokenization process. Additionally, it is necessary to elaborate on what constitutes the resulting sets used for computing precision and recall. Providing details on the tokenization process, including any specific algorithms or techniques utilized, would enhance understanding and reproducibility of the evaluation methodology.

**Detailed Comments:**

N/A

**Justification Of Final Rating:**

Thank you to the authors for addressing my doubts in the rebuttal.
The authors claim they have thoroughly verified the correctness of their implementation via four aspects. They also corrected the implementation of RadFM and LLaVA-Med and updated the evaluation results.
All of my concerns have been addressed.

**Justification Of The Preliminary Rating:**

The toolkit proposed in this paper offers a unified benchmark for VLMs, effectively tackling the issue of non-uniformity in current evaluation methodologies. However, upon a thorough examination of the performance table generated by MultiMedEval, doubts regarding credibility and reliability have emerged, leading to a change in the rating from accept to borderline. Providing an explanation in the rebuttal is imperative to address these concerns.

**Questions To Address In The Rebuttal:**

In the rebuttal, the authors must elucidate the reasons behind the poor performance scores assessed by MultiMedEval. Additionally, further evidence is required to bolster the reliability of the proposed toolkit. Furthermore, a detailed clarification of the evaluation methodology for Visual Question Answering (VQA) is essential.

**Special Issue:**

No

---

> ### Author Response · Authors · 2024-03-16
>
> Thank you for your review and for appreciating the scope and goals of our work. We would like to address the three concerns you expressed and we kindly ask you to look at the revised version of our manuscript.
>
>
> ## Concern 1: Accuracy of the evaluation
>
>
> The accuracy and reliability of the benchmark are paramount. To ensure this, we provide assurances for the 4 potential sources of error: (A) data download, (B) prompt construction, (C) raw answer processing, and (D) metric computation.
>
>
> For (A), (C), and (D) we implemented unit tests and a continuous integration process in the GitHub repository. Specifically, we have:
> * Correctness of the data download ([Unit test file](https://github.com/corentin-ryr/MultiMedEval/blob/main/tests/test_loading_all.py))
> * Correctness of the tokenizer for VQA tasks ([Unit test file](https://github.com/corentin-ryr/MultiMedEval/blob/main/tests/test_vqa_preprocessing.py))
> * Accuracy of compound metrics such as RadCliQ ([Unit test file](https://github.com/corentin-ryr/MultiMedEval/blob/main/tests/test_report_comparison.py)). Simpler metrics like accuracy and F1 score are taken from libraries like sklearn and pytorch.
>
> As for the source of error (B), we base our prompt construction on previous work whenever possible and construct a suitable prompt for every other case. We provide a new table (Table A7) in the appendix of the revised manuscript to clarify this process. Our goal is to propose a standard prompt for each task to minimize the disparities due to prompt sensitivity (a problem acknowledged by previous works [RadFM issue](https://github.com/chaoyi-wu/RadFM/issues/10))
>
>
> ## Concern 2: Result discrepancies
>
> The low numbers in Table 4 (image classification and NLI) are simply explained by the absence of these modalities in the training data of RadFM and LLaVA-Med. For example, the image classification tasks cover a large range of different modalities (dermatology, pathology...) whereas RadFM was only trained on radiology tasks leading to wrong or even invalid answers and low performance in these tasks.
>
> We want to support MultiMedEval in the same open-source spirit as similar to tools like [SentEval](https://github.com/facebookresearch/SentEval/issues). Thanks to your feedback, we realized that there was a discrepancy in the report generation task. This was due to the prompting setup that we used compared to the one in the other papers. Specifically, the implementation of this benchmark in RadFM and MedPaLM M included the indication section of the report in the prompt and we only included the images. We addressed this issue and reevaluated RadFM and LLaVA-Med. You can find the updated results below and in the revision of our manuscript.
>
>
> |Dataset|Metric|RadFM|RaFM(reported)|LLaVA-Med|LLaVA-Med(reported)|
> |---------|---------------|-----|---------------|---------|---------|
> |MIMIC-III|BLEU-1|0.127|\-|0.129|\-|
> ||BLEU-4|0.016|\-|0.030|\-|
> ||ROUGE-1|0.221|\-|0.222|\-|
> ||ROUGE-L|0.157|\-|0.164|\-|
> ||F1-RadGraph|0.149|\-|0.211|\-|
> ||RadCliQ|1.396|\-|1.315|\-|
> ||CheXbertvector|0.602|\-|0.596|\-|
> ||METEOR|0.191|\-|0.314|\-|
> |MIMIC-CXRReportGeneration|BLEU-1|0.127|0.194|0.109|\-|
> ||BLEU-4|0.004|\-|0.000|\-|
> ||ROUGE-1|0.208|0.262|0.149|\-|
> ||ROUGE-L|0.128|\-|0.103|\-|
> ||F1-RadGraph|0.108|\-|0.021|\-|
> ||RadCliQ|1.994|\-|2.415|\-|
> ||CheXbertvector|0.214|\-|0.156|\-|
> ||METEOR|0.178|\-|0.123|\-|
>
>
> The difference between our numbers and the reported numbers is also explained by the stochastic generation method (sampling from the token distribution) which we adopted from the original RadFM and LLaVA-Med implementation.
>
> We would like to stress that successful reproduction in MultiMedEval indicates that the metric computation is accurate. However, RadFM has not yet made its evaluation code publicly available making it difficult to pinpoint the exact differences. Once released, we will examine the code and incorporate any necessary improvements, thereby emphasizing the importance of community consensus.
>
>
> ## Concern 3: VQA tokenization
>
> In the interest of conciseness, we did not go in-depth into the details of the tokenization and in light of your feedback on the performance discrepancy, we decided to align our tokenization procedure with the one used by LLaVA-Med. The updated procedure consists of first cleaning the generated string by removing punctuation, articles ("a", "an", "the"), turning all numbers into numerals ("one" -> "1"), and expanding contractions that lack apostrophes. We then split the sentence at spaces and end up with a list of words that we turn into a set (removing duplicates). After these changes, we reproduced LLaVa-Med's performance, as shown above in the table (also included in the revised manuscript), and we added the tokenization details in the appendix section of the manuscript.

---

### Official Review · Reviewer_4Kwo · 2024-02-28

**Confidence:** 3
**Preliminary Rating:** 2
**Final Rating:** 3.5

**Summary:**

The paper introduces MultiMedEval, a Python-based evaluation toolkit designed for medical VLMs. MultiMedEval provides reproducible evaluation functions for six different medical tasks, applied to 23 datasets covering 11 medical image and text modalities.

**Strengths:**

The motivation is evident and the idea for creating a unified evaluation platform is critical and necessary.

The paper is well-structured, and the writing is concise.

The range of medical tasks supported by the Toolkit is extensive.

**Weaknesses:**

The primary concern raised by the reviewer pertains to the accuracy of implementation, as depicted in Tables 4 and 5.

Whether the Toolkit is user-friendly and easy to use is another concern from reviewers.

**Detailed Comments:**

The paper discusses the re-implementation of two open-source models (RadFM and LLAVA-Med). However, as shown in the benchmark results in Table 5, the re-implemented models consistently exhibit lower performance than the reported values. This raises concerns regarding the accuracy of the implementation.


In Table 4, the performance of the two re-implemented models is notably inferior (0.001 accuracy? ) compared to other models. This discrepancy may lead potential users to question the accuracy of the evaluation.


The tool would be enhanced in capability with the inclusion of a data pre-processing module.

**Justification Of Final Rating:**

Considering that the majority of my concerns have been adequately addressed and recognizing the positive potential of the proposed framework, I am inclined to adjust my final rating to "Borderline accept."

**Justification Of The Preliminary Rating:**

The paper presents MultiMedEval, an evaluation toolkit tailored for medical VLMs. The motivation behind developing this toolkit is substantial. However, as a toolkit, the paramount considerations are the accuracy of implementation and operational efficiency, aspects that are not adequately demonstrated in the paper. As a result, we give a weak reject rating for the preliminary assessment.

**Questions To Address In The Rebuttal:**

The concerns from the above comments should be carefully addressed.

---

> ### Author Response · Authors · 2024-03-16
>
> Thank you for appreciating the need for this benchmark and especially for identifying the two areas where we could improve tremendously. Below we address your three major concerns and are happy to hear back from you.
>
> ### Concern 1: Accuracy of the evaluation
>
> We aim to address the broader concern regarding the overall accuracy of our implementation. MultiMedEval encompasses several tasks, each with potential for errors: data download, prompt construction, VLM response decoding, and metric computation. Specifically, the following errors are possible: (A) incorrect data download, (B) inaccurate model prompting, (C) incorrect decoding of the VLM's response, and (D) incorrect metric computation.
>
> To ensure the validity of the evaluation and address errors A, C, and D, we implemented a suite of unit tests and a continuous integration pipeline. These tests verify the following:
>
> * Correctness of the data download ([Unit test file](https://github.com/corentin-ryr/MultiMedEval/blob/main/tests/test_loading_all.py))
> * Correctness of the tokenizer for VQA ([Unit test file](https://github.com/corentin-ryr/MultiMedEval/blob/main/tests/test_vqa_preprocessing.py))
> * Accuracy of compound metrics such as RadCliQ ([Unit test file](https://github.com/corentin-ryr/MultiMedEval/blob/main/tests/test_report_comparison.py)). Simpler metrics like accuracy and F1 score are taken from libraries like sklearn and pytorch.
>
> Regarding error B (input prompt construction), MultiMedEval uses prompts from prior work whenever possible. We provide this information in a new table (Table A7) in the supplementary material of the revised manuscript. In cases where the prompt is modified, it is due to disagreements in the community. MultiMedEval proposes using an adapted version as the future standard to mitigate the well-known concern surrounding VLMs' sensitivity to input prompts [RadFM issue](https://github.com/chaoyi-wu/RadFM/issues/10).
>
> ### Concern 2: Re-implemented models consistently exhibit lower performance than the reported values.
>
> Language generation is probabilistic, and additional stochasticity is introduced by previous work. For example, during inference, models typically set sampling=True for token prediction. Non-standard prompting, as mentioned earlier, also introduces randomness. RadFM, for example, randomly selects a prompt for report generation ([RadFM report generation prompts](https://github.com/chaoyi-wu/RadFM/blob/main/src/Dataset/dataset/report_prompt.json)). We have attempted to minimize this stochasticity, which contributes to discrepancies.
>
> In line with open-source principles, MultiMedEval will continue to improve based on user feedback, similar to tools like SentEval (https://github.com/facebookresearch/SentEval/issues). Your feedback helped us identify a discrepancy in text normalization between LLaVa-Med and MultiMedEval, with the latter using a more generic routine. After updating the routine, we reproduced LLaVa-Med's performance, as shown below (also included in the revised manuscript in Tables 4 and 5).
>
> |Dataset|Metric|RadFM|RaFM(reported)|LLaVA-Med|LLaVA-Med(reported)|
> |-|-|--|-|-|--|
> |VQA-Rad|BLEU-1|0.443|0.522|0.058|\-|
> ||closedQaccuracy|0.581|\-|0.621|0.614|
> ||openQrecall|0.326|\-|0.335|0.282|
> ||recall|0.468|0.428|0.485|\-|
> ||openQaccuracy|0.260|\-|0.255|\-|
> ||F1|0.454|\-|0.102|\-|
> |SLAKE|BLEU-1|0.707|0.786|0.051|\-|
> ||closedQaccuracy|0.715|\-|0.515|0.522|
> ||openQrecall|0.720|\-|0.408|0.392|
> ||recall|0.718|0.744|0.444|\-|
> ||openQaccuracy|0.684|\-|0.361|\-|
> ||F1|0.718|\-|0.090|\-|
> |VQA-Path|BLEU-1|0.254|\-|0.029|\-|
> ||closedQaccuracy|0.500|\-|0.511|0.541|
> ||openQrecall|0.019|\-|0.079|0.123|
> ||recall|0.259|\-|0.290|\-|
> ||openQaccuracy|0.008|\-|0.037|\-|
> ||F1|0.256|\-|0.052|\-|
>
> It is important to note that successful reproduction here indicates that the metric computation is accurate. However, RadFM has not yet made its evaluation code publicly available. Once released, we will examine the differences and incorporate any necessary improvements, thereby emphasizing the importance of community consensus.
>
> ### Other concerns
> *Poor performance on MedNLI*: The metric used for MedNLI has been thoroughly unit-tested. The suboptimal performance can be primarily attributed to the current limitations of VLM's capability in language inference. With the introduction of MultiMedEval, we encourage future VLMs to also assess their performance on LLM tasks.
>
> *Ease of Use*: We respectfully differ from the reviewer's concern regarding the ease of use. The most significant challenges in evaluating VLMs lie in consolidating the datasets and implementing the performance metrics. This is handled seamlessly in the background, simplifying VLM evaluation into a few lines of code. Suggestions on improving the ease of use are very welcome.
>
> *Data Pre-processing*: Data pre-processing is more of a model-specific requirement than a dataset-specific one. MultiMedEval only handles dataset-side processing, such as file system operations.

---

### Author Response · Authors · 2024-03-16

Thank you for taking the time to review our paper and for providing valuable feedback. Based on your insights we made some adjustments to our work. Here's a summary of the changes we made based on your suggestions:
## Accuracy of Evaluation:
We have thoroughly addressed concerns regarding potential errors in our implementation, including data download, prompt construction, VLM response decoding, and metric computation. To ensure accuracy, we implemented a suite of unit tests and a continuous integration pipeline, and we added more details on our tokenization process. We also gave more details on our prompt construction approach and provided a new table in the supplementary material to clarify this process.

## Result Discrepancies:
We clarified the origin of the discrepancies between our results and the reported ones, particularly regarding image classification and report generation. By aligning our tokenization procedure with previous works, we have achieved more consistent and reliable performance metrics. Additionally, we await the release of RadFM's evaluation code to further ensure alignment and foster community consensus.

## Incorporation of Diff-VQA and Flexibility:
We have integrated Diff-VQA into MultiMedEval, acknowledging the importance of accommodating diverse datasets. Additionally, we have introduced a new feature allowing users to easily add any new dataset to MultiMedEval, streamlining the benchmarking process and enhancing its flexibility.

## Fairness:
In response to concerns raised about fairness in evaluation, we are actively exploring options for more detailed, stratified analysis per task. This includes considerations for evaluating datasets such as Pad UFES 20 and potentially incorporating additional datasets like MIMIC and Genderbias_CheXnet. We recognize the importance of fairness in benchmarking, especially in the context of medical applications where biases can have significant implications.


We would greatly appreciate any further feedback you may have, particularly on the revised manuscript. Your insights are valued in our efforts to improve MultiMedEval and ensure its effectiveness as a benchmarking tool.

---

> ### Author Response · Authors · 2024-03-25
>
> Dear Reviewers,
>
> Thank you for your valuable feedback on our submission. As you saw, we addressed the concerns you raised and made significant improvements to our work. We are open to further discussion and collaboration to ensure the robustness and effectiveness of MultiMedEval as a benchmarking tool.
>
> Please feel free to engage with us during the discussion period and we will do our best to provide further details.

---

### Meta-Review · Area_Chair_a9tx · 2024-04-03

**Recommendation:** Accept (Poster)
**Confidence:** 5

**Metareview:**

The authors have addressed the majority concerns of reviewers after rebuttal. The reviewers all confirm the merits of this paper (1 Borderline accept, 1 Weak accept and 1 Strong accept after rebuttal). Given the consensus of reviewers, a decision of accept is recommended.

---

### Decision · Program_Chairs · 2024-04-05

Accept (Poster)